# Anomalous size effect on yield strength enabled by compositional heterogeneity in high-entropy alloy nanoparticles

Jingyuan Yan[1,5], Sheng Yin[2], Mark Asta [2,3], Robert O. Ritchie [2,3], Jun Ding [4✉] & Qian Yu [1✉]

High-entropy alloys (HEAs), although often presumed to be random solid solutions, have recently been shown to display nanometer-scale variations in the arrangements of their multiple chemical elements. Here, we study the effects of this compositional heterogeneity in HEAs on their mechanical properties using in situ compression testing in the transmission electron microscope (TEM), combined with molecular dynamics simulations. We report an anomalous size effect on the yield strength in HEAs, arising from such compositional heterogeneity. By progressively reducing the sample size, HEAs initially display the classical "smaller-is-stronger" phenomenon, similar to pure metals and conventional alloys. However, as the sample size is decreased below a critical characteristic length (~180 nm), influenced by the size-scale of compositional heterogeneity, a transition from homogeneous deformation to a heterogeneous distribution of planar slip is observed, coupled with an anomalous "smaller-is-weaker" size effect. Atomic-scale computational modeling shows these observations arise due to compositional fluctuations over a few nanometers. These results demonstrate the efficacy of influencing mechanical properties in HEAs through control of local compositional variations at the nanoscale.

[1] Center of Electron Microscopy and State Key Laboratory of Silicon Materials, Department of Materials Science and Engineering, Zhejiang University, 310027 Hangzhou, China. [2] Materials Sciences Division, Lawrence Berkeley National Laboratory Berkeley, Berkeley, CA 94720, USA. [3] Department of Materials Science and Engineering, University of California, Berkeley, CA 94720, USA. [4] Center for Alloy Innovation and Design (CAID), State Key Laboratory for Mechanical Behavior of Materials, Xi'an Jiaotong University, Xi'an 710049, China. [5] Present address: Institute of Engineering Innovation, The University of Tokyo, Tokyo 113-8656, Japan. ✉email: dingsn@xjtu.edu.cn; yu_qian@zju.edu.cn

High-entropy alloys (HEAs) have been the subject of intensive research over the past decade[1–8]. Such multiple principal element alloys can form concentrated single-phase solid solutions, which often display, particularly in the case of the face-centered cubic (fcc) CrCoNi-based alloys, exceptional mechanical properties[9,10]. HEA solid solutions were originally thought to involve a nominally random chemical arrangement of different atoms over the sites of the underlying parent lattice[1,11]. However, recently, both experimental and computational studies have shown evidence of the existence of non-random distributions of the elemental constituents. Extended x-ray absorption fine structure (EXAFS) studies were first used to show that the Cr in CrCoNi tends to bond with Ni and Co to form short-range order[12]. More recently, both atomic-resolution energy-dispersive x-ray spectroscopy (EDS) and energy-filtered transmission electron microscopy have enabled the direct observation of such local chemical heterogeneity[13–15]. Additionally, models considering such heterogeneity have been found to be more appropriate to explain the pair distribution function measured in neutron scattering[16]. Several studies involving atomistic simulations have also revealed that local chemical ordering heightens the ruggedness of energy landscape and raises activation barriers for dislocation activity in fcc alloys[12,17]. Specifically, the nature of the non-random distributions of the elemental constituents in HEAs have been detected in the form of either chemical short-range-order (CSRO)[14] or variations in local composition over longer length-scales[13].

Theories based on random solid solutions have been developed for the strengthening of HEAs from dislocation-based plasticity, and successfully applied for predicting the yield strength of several HEA systems[18–20]. However, additional observations, such as the difference of the exponent $n$ in the traditional size-effect relationship, $\sigma = \sigma_0 + kd^{-n}$ ($\sigma$ is the yield stress, $\sigma_0$ is the friction stress, i.e., the lattice resistance to dislocation movement, $d$ is the sample size and $n$ is the exponent factor which is usually ~½) between HEAs and conventional metals and alloys[21,22], and fluctuations in lattice friction and local stacking-fault energy (SFE)[17,23–25] have motivated complementary models considering the effect of compositional heterogeneity in regulating dislocation activity and deformation. As the first crucial step to fully understand the structure-property relationships in these HEA concentrated solid solutions, it is important to define all the salient mechanisms at play, especially the role of compositional heterogeneity, to characterize them in physical terms.

Here, by CSRO we refer to the preference for certain types of unlike chemical bonds over a few neighbor shells, (i.e., on the scale of a lattice parameter), in a manner consistent with the preference for forming ordered intermetallic compounds. In comparison, clustering or segregation causes regions of different composition to form on longer length scales (e.g., several nanometers), in a manner consistent with systems displaying miscibility gaps. These two cases both describe the compositional heterogeneity from local "ordering" to "clustering", with the associated length-scale varying from Ångstroms to tens of nanometers or more, respectively. In the present work, we specifically investigate the existence and role on mechanical deformation associated with such structural ordering and clustering phenomena, which we will collectively refer to as "compositional heterogeneity" at different length scales in face-centered cubic (fcc) HEAs.

To achieve this important understanding, a clean experimental system is required where compositional heterogeneity is known to be prominent, ideally as the major microstructural characteristic influencing the mechanical behavior of the material. However, it is worth noting here that at the bulk scale, the macroscopic mechanical properties and deformation modes represent the integrated average of innumerable different strengthening mechanisms that are influenced by boundary and orientation effects; as such, detecting the presence of compositional heterogeneity and its specific effect on mechanical properties, as compared to that due to a random solid solution, can be difficult to isolate. In light of these issues, our approach here is to focus on small-scale HEAs, at size dimensions similar to that of the length-scale of the compositional heterogeneity, so that intrinsic effects arising from the presence of such compositional heterogeneity would not be masked by other phenomena. We believe that the insights derived from this work will provide an essential quantification of the important role of compositional heterogeneity in influencing the deformation behavior of HEAs, specifically with respect to the activation of unique combinations of deformation modes and the mechanisms that govern the yield strength and strain hardening[25–27].

For pure metals and conventional alloys, the size effect of the yield strength follows the widely acknowledged "smaller-is-stronger" relationship which, provided the chemical distribution is homogeneous and random (such that the sample size is the only variable), takes the form of $\sigma = \sigma_0 + kd^{-n}$. However, with the presence of compositional heterogeneity, the value of $\sigma_0$ can no longer be deemed to be a constant during the decrease in sample size. This is because such heterogeneity would be expected to have an influence on local lattice resistance and thus result in local variation of $\sigma_0$ when the sample size is reduced to dimensions comparable to the spatial correlation length of the prevalent compositional heterogeneity[17]. The critical size for such a transition from a normal to an abnormal size effect is termed $L_{critical}$, and is presumed to relate to the spatial correlation length of compositional heterogeneity, $L_{hetero}$. To date, most reports[26–30] of microscale mechanical properties that focus on the size effect in HEAs have only involved alloys with combinations of elements that are close in the Periodic Table, such as iron, cobalt, nickel, chromium, manganese. In these alloys, compositional heterogeneity may exist but is presumably much smaller in length-scale[13] compared to the sample sizes, i.e., $L_{hetero} \ll d$. Research on HEAs to date has involved sample sizes ranging from several micrometers to hundreds of nanometers; currently, the smallest sample sizes are on the order of 200 nm, as used in indentation experiments and for the compression of pillars of fcc HEAs, but this is roughly two orders of magnitude larger than the length-scale of any compositional heterogeneity which would be in the range of <0.5 to ~10 nm[12–15]. Indeed, numerous studies to date, using sample dimensions in HEAs in the hundreds of nanometers, i.e., far larger than $L_{critical}$, still obey the usual "smaller-is-stronger" phenomenon[21,22,26,27,31,32].

In this work, we attempt to investigate the effect and working size range of compositional heterogeneity on the mechanical properties, specifically the yield stress and work-hardening ability, by quantifying the size effect of HEA nanoparticles by in situ nano-compression testing on single crystals. As noted in our previous research, a mixture of elements with a large difference in electronegativity and atomic size would be expected to generate stronger heterogeneity in the distribution of elements[12], thereby resulting in an increased $L_{critical}$ and hence providing the possibility to experimentally study the influence of the local heterogeneity on the mechanical properties, i.e., at sample sizes of $d$ at the same dimension of $L_{critical}$. In this study, $Fe_{18}Co_{16}Ni_8Cu_{20}Pd_{12.5}Ir_{1.5}Pt_{10}Au_{14}$ nanoparticles (see Fig. 1a) were utilized for in situ compression tests. Approximately 80 nm to 300 nm sized particles were mechanically tested using nano-compression loading along crystallographic directions within 15 to 30 degrees of <111> inside an FEI Tecnai G2 F20 transmission electron microscope equipped with a PI95 nanoindenter. We find that as the sample size decreases below the $L_{critical}$ of ~ 180 nm, an anomalous "smaller is weaker" size effect replaces the traditional "smaller is stronger" trend, accompanied with the transition from

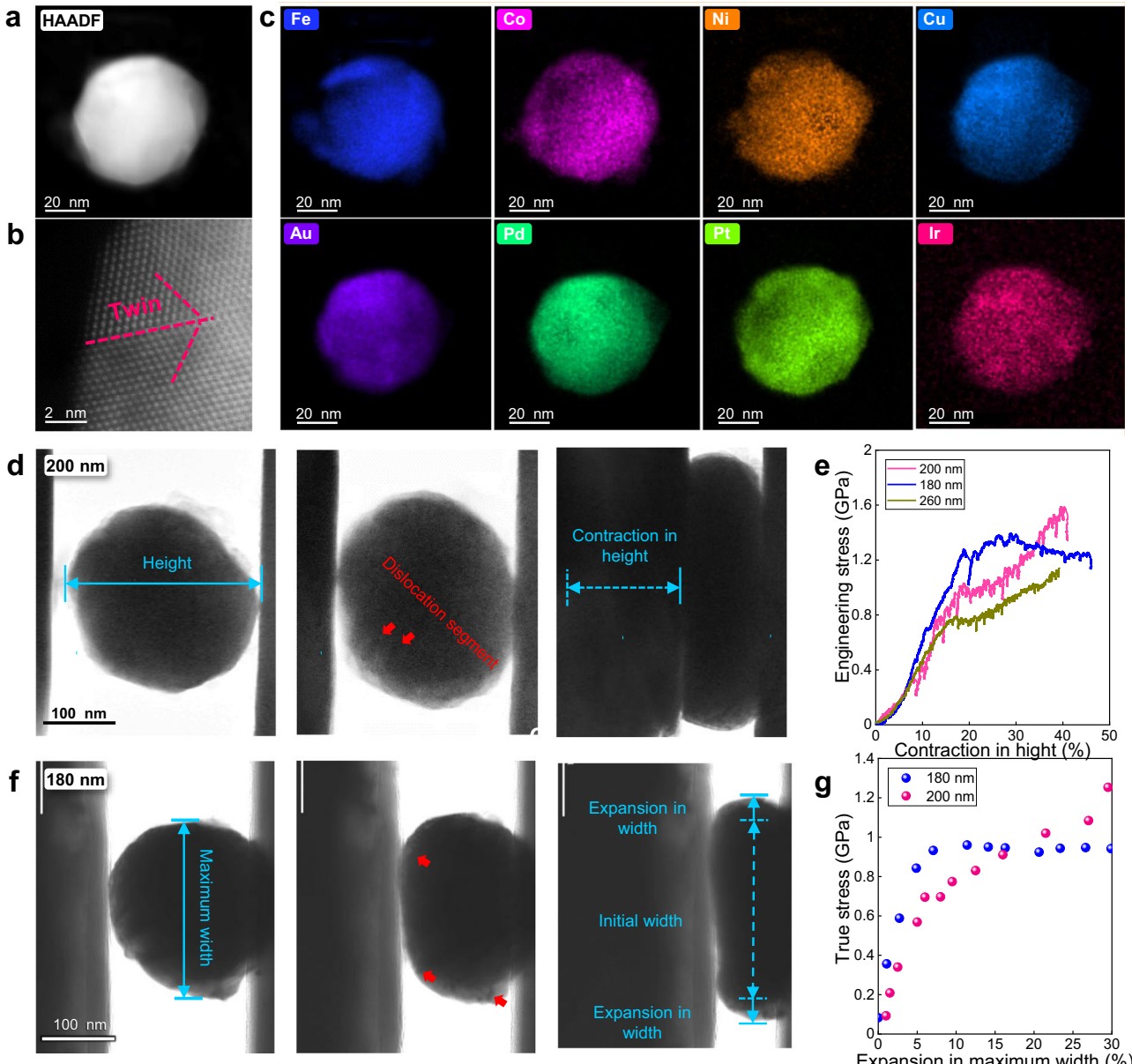

**Fig. 1 Characterization of the FeCoNiCuPdIrPtAu nanoparticles and the "smaller-is-stronger" phenomenon in particles with diameter larger than 180 nm. a** HADDF image showing the morphology of a particle with diameters of 50 nm. **b** HADDF-STEM image showing the atomic structure of a particle containing a twin (beam direction [110]). **c** EDS mapping results showing how the eight elements are distributed in the particles. Cu, Au, and Pd display a similar distribution that is complementary to that of Fe, Co, Ni and Pt; note that there is no significant segregation of noble elements at the surface of the nanoparticles, and compositional variations occur within the nanoparticles (also see Supplementary Fig. 1) **d, f** Deformation of large particles with a diameter of 200 nm and 180 nm, respectively (frames are taken from Supplementary Movie 1 and 2); the centers of the particles were barely electron transparent due to their comparatively large size and because their composition comprised certain noble elements with large atomic numbers.
**e** Engineering stress *vs.* contraction-in-height curves showing the "smaller-is-stronger" trend, based on tests on the particles in (**d, f**) and a larger 260 nm sized particle. The engineering stress was calculated from the continuously recorded load values and diameters calculated with the cylinder model (see Methods for details)). **g** True stress *vs.* expansion-in-maximum width diagrams of (**d, f**) (stresses were calculated from the real-time load values and diameters measured from the extracted frames).

a homogeneous "flow-like" deformation behavior to an instable one characterized by dislocation avalanche and catastrophic localized shear. Further computational simulation results also provide evidence that the compositional heterogeneity would change the distribution of the local stacking fault energies (SFEs) and therefore modify the deformation behavior and mechanical properties of HEAs.

## Results

**Microstructure and element distribution**. Figure 1a shows a typical HAADF image of the $Fe_{18}Co_{16}Ni_8Cu_{20}Pd_{12.5}Ir_{1.5}Pt_{10}Au_{14}$ nanoparticles. The particles were *fcc* structured spherical single crystals although some of them contained twins, as shown in Fig. 1b. The EDS results in Fig. 1c suggest that element clustering prevails over spatial dimensions up to 10 nm.

**The traditional "smaller-is-stronger" trend above $L_{critical}$.** Figure 1d, f shows the nature of the deformation in the 200 and 180 nm sized FeCoNiCuPdIrPtAu nanoparticles, respectively. Both particles assumed a shape evolution through large-scale plastic deformation, being compressed into a perfect "pie shape" with expansion in maximum width of over 30%. This occurred without obvious evidence of shear or cracking which has been reported for many other *fcc* pure metals and HEAs at larger scales[26,27,33,34]. The resulting engineering stress *vs.* reduction-in-height curves can be seen to be relatively smooth (Fig. 1e). The yield strengths of the 260, 200, and 180 nm particles were, respectively, 0.78, 1.0, and 1.25 GPa, i.e., they displayed the traditional "smaller-is-stronger" trend. The 260 and 200 nm sized particles, in particular, showed strong strain-hardening ability, with engineering stresses at 40% contraction-in-height that were ~50% higher than that at their yield points; the 180 nm sized nanoparticle conversely exhibited much reduced strain hardening after yielding, indicating a change in the energy barrier for dislocation movement. In view of the lack of accuracy in the quoted stresses introduced by simple "engineering stress" calculations, true stresses were computed from the real-time load values and diameters measured from extracted frames; resulting true stress *vs.* expansion-in-maximum-width diagrams are presented in Fig. 1g. The stress values are slightly different but the "smaller-is-stronger" trend remains the same.

**The anomalous "smaller-is-stronger" trend below $L_{critical}$.** Interestingly, as the particle sizes were further decreased, the yield stress no longer increases and an anomalous "smaller-is-weaker" phenomenon becomes apparent. Additionally, the deformation mode was significantly altered. As shown in Fig. 2, plastic instability was in evidence during the deformation of the 140 nm

sized and 80 nm sized particles; a transition in the mode of plasticity from homogeneous ("flow-like") deformation to dislocation avalanche and catastrophic localized shear occurred in these smaller sized particles. As shown in Fig. 2a, b, shear localization and rapid dislocation movement were observed during compression (shown in Supplementary Movies 3, 4), which also indicates that the deformation processes were not dominated by surface diffusion[35,36]. In addition, the mechanical deformation curves shown in Fig. 2c were more serrated, accompanied by marked and random load drops, which were not observed for the larger particles, with minimal strain hardening (Fig. 2c, d). The yield strength of the ~140 and ~80 nm particles were, respectively, 1.08 and 0.8 GPa, showing a "smaller-is-weaker" trend.

Since it is difficult to perform compression tests on nanoparticles with even smaller sizes, we further tested the mechanical behavior of some FeCoNiCuPdIrPtAu nanoparticles containing twins in which the twin boundaries were parallel to the slip plane. This effectively divided such particles into smaller parallel units with strain states comparable to the experiments for the single crystals described above, while the size of the slip plane, and thus the distance of dislocation glide, is confined by the twin boundaries and is similar to that in a smaller particle. Thus, the dislocation motion within each grain is essentially independent and can be deemed as the mechanical response in smaller particles. Figure 3 shows such a particle, 140 nm in diameter, with three neighboring twins inside. Each unit was smaller than 50 nm in width. As shown in the mechanical data, it was found that this particle displayed even lower yield strength (~0.5 GPa) as compared to single grain particles of similar size and even the 80 nm sized particles, as shown in Fig. 3a, b.

The transition of the "smaller-is-stronger" to the "smaller-is-weaker" phenomena with the reduction in particle size together with the change in work-hardening ability are presented in

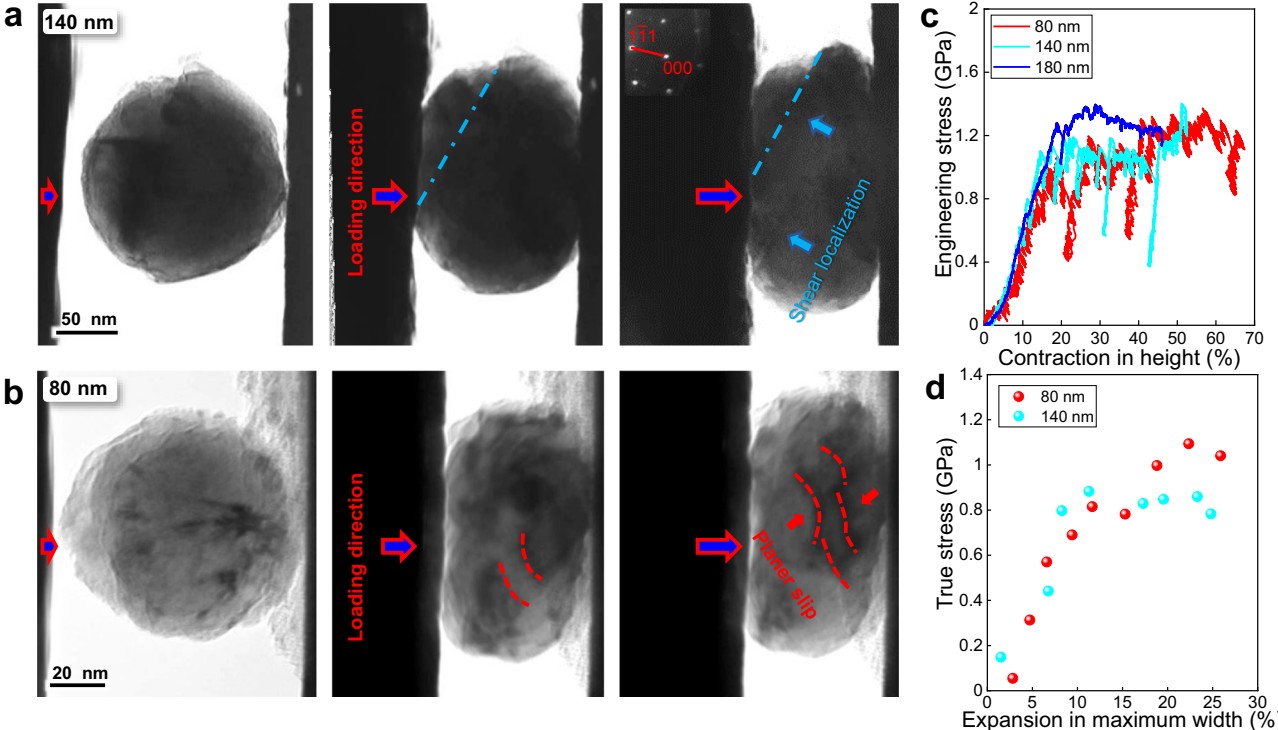

**Fig. 2 Unstable deformation behavior in smaller FeCoNiCuPdIrPtAu nanoparticles which display a "smaller-is-weaker" phenomenon.** Deformation process (**a**) of a particle with a diameter of 140 nm, and (**b**) with diameter of 80 nm. **c** Engineering stress *vs.* reduction-in-height curves for the 80 nm and 140 nm smaller particles as compared with that for the 180 nm sized particles. **d** Representative true stress *vs.* expansion-in-maximum-width curves for the 80 and 140 nm sized particles.

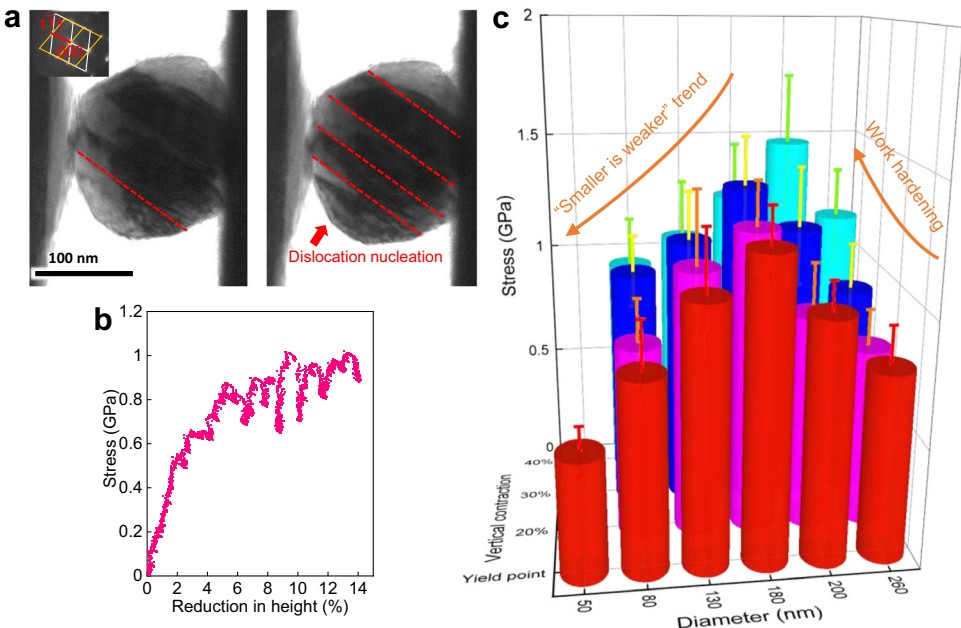

**Fig. 3 The "smaller-is-weaker" relationship further confirmed by particles divided with twins. a** The yield process of a FeCoNiCuPdIrPtAu nanoparticle with diameter of 140 nm with twin structures inside, which divide it to smaller units with a width below 50 nm (shown in Supplementary Movie 5). **b** The engineering stress *vs.* reduction-in-height curve corresponding to the deformation process in (**a**). **c** Analysis of strengths for different particles at specific values of their reduction-in-height. Each presented value is averaged over results for more than three particles. The deviation is within 15% in particles larger than 180 nm in diameter but increases to about 33% in particles smaller than 180 nm in diameter, as described by the error bar.

**Table 1 The yield stress and hardening percentage of different sized samples.**

| Particle diameter | Yield stress | Strain hardening ratio (at 40% strain) |
|---|---|---|
| 80 nm | 0.8 GPa | 18.7 % |
| 140 nm | 1.08 GPa | −1 % |
| 180 nm | 1.25 GPa | 0 % |
| 200 nm | 1 GPa | 48 % |
| 260 nm | 0.78 GPa | 47.4 % |

Fig. 3c. Each column represents the averaged stress value at certain vertical contraction for all the nanoparticles tested at a similar size. For the same degree of deformation, the stresses decrease as the particles size is reduced below ∼ 180 nm in the form of a "smaller-is-weaker" trend. Additionally, the work-hardening ability drastically deteriorates, from ∼ 50% to around 0% (shown in Table 1), after the "smaller-is-stronger" to "smaller is weaker" transition, indicating that dislocation interactions might be more extensive in the larger particles.

It should be noted that the particles in our research are all single crystals, which eliminated any effect of grain boundaries that have been previously reported to result a "smaller is weaker" phenomena[37–39]. Further, the melting temperature of the particle exceeds 1000 K[40], whereas our calculation (provided in Supplementary Note 1) shows that the temperature rise during the TEM observation is quite small and the particle remains at ambient temperature during the in situ compression tests, which is consistent with previous reports[41,42]. Specifically, any electron beam heating effect, which could result in a "surface-diffusion" dominated "smaller-is-weaker" trend, is negligible[35,36].

To summarize these results, with progressively decreasing size, the strength of our eight-element HEA nanoparticles exhibited a transition from the "smaller-is-stronger" to "smaller-is-weaker" phenomenon below a critical size $L_{critical}$ of ∼180 nm; additionally,

below this size, the deformation of the particles started to display a concomitant loss in a capability for dislocation storage. The "smaller-is-stronger" size effect has been widely reported in traditional pure metals and alloys[28–32] where the resistance of the lattice to dislocation motion, i.e., the friction stress term $\sigma_0$ in the traditional size-effect relationship, $\sigma = \sigma_0 + kd^{-n}$, is a constant; this is an important prerequisite for the "smaller-is-stronger" phenomenon. The strength increases as the sample size is reduced to the submicrometer or nanometer scale because the length of the dislocation lines $L$ becomes determined by the sample size, i.e., $\sigma \sim L^{-n} \sim d^{-n}$ (where $\sigma$ is strength, $L$ is the length of the dislocations and $d$ is the sample size)[35]. However, such a "truncation mechanism"[43,44], which has been utilized to explain behavior in traditional metals and alloys, is not sufficient when it comes to HEAs because the compositional heterogeneity can result in additional effects to alter the mechanical properties, but which are masked by other mechanisms at larger size scales.

**Simulated "smaller-is-weaker" size effect in CrCoNi.** To derive further insights into the origins of the observed anomalous size effect and its relationship with compositional heterogeneity in this HEA, we employed atomistic simulations[45] to discern the transition of the size effect on the strength of HEAs, in comparison to that in pure metals and conventional alloys. Since realistic force-field models for the eight-element alloy system are not available, MD simulations were performed utilizing a recently developed empirical EAM potential for modeling CrCoNi medium-entropy alloys[17]. A number of [110] oriented CrCoNi rhombic nanowires, either with random solid solutions (RSS) or with configurations resulting from annealing at 600 K (with enhanced compositional heterogeneity), shown respectively in Fig. 4a, were initially constructed for tensile tests by MD simulation, with close-packed {111} side surfaces; this is described in detail in the Methods section. Pure Ni rhombic nanowires with no compositional heterogeneity were also simulated for comparison. Figure 4b shows the example atomic configurations of a

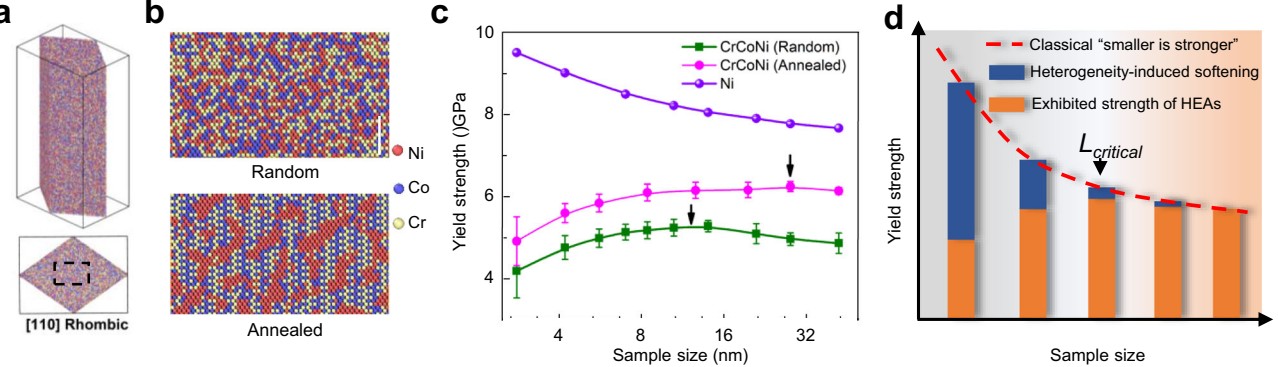

**Fig. 4 MD simulation of size-dependent yield strength. a** The side view and top view, respectively, of a [110] oriented rhombic nanowire of a CrCoNi alloy with {111} side surfaces (colored according to atomic species). **b** The example configurations of {110} surface in random solid solution and annealed CrCoNi alloy, respectively; (scale bar 2 nm). **c** The yield stress versus sample sizes for the nanowires of the CrCoNi alloy with random solid solution and annealing (i.e., at 600 K), as well as pure Ni (the error bars show the deviation of the results around the averaged values). The sample size was defined as the length of the larger diagonal. **d** The schematic description for the size effect in high-entropy alloys, combining the classical strengthening effect for small-volume metals/alloys and the heterogeneity-induced softening for HEAs.

{110} surface for the random solid solution and annealed CrCoNi alloys; the former displays a random distribution of the three elements, whereas clustering or segregation at the length scale of ~1.5 nm (as well as chemical short-range order) can be observed in the latter samples. A number (5–500) of these nanowires was chosen for each size to ensure converged yield stress values in the simulations. Figure 4c shows plots of the average yield stress versus sample size for each material. Nickel can be seen to exhibit an obvious "smaller-is-stronger" trend that is consistent with the results on copper[45], while the random solid solution CrCoNi alloys and the annealed samples show a contrasting sample size effect with a critical size, $L_{critical}$, of ~12 nm and ~30 nm, respectively, transitioning from "smaller-is-stronger" to "smaller-is-weaker" with decreasing sample size.

## Discussion

Based on the above information, we propose here a qualitative model to explain how this anomalous size effect arises from compositional heterogeneity in HEAs. As schematically illustrated in Fig. 4d, a smaller sample size, on the one hand, contributes to strengthening following the classical "smaller-is-stronger" relationship above a certain dimension; on the other hand, it raises the variation in lattice friction among parallel atom planes, since the averaging over the different possible local environments on each plane is altered due to the limited area of the plane. We believe that these two competing factors acting synergistically result in the characteristic $L_{critical}$ in nanoscale HEA nanoparticles observed in this research. Specifically, deformation becomes altered by the heterogeneous nature of the structure over the length-scale of the sample size, resulting in the opposite "smaller-is-weaker" size effect (as discussed in detail below). The competition between these two opposing mechanisms results in the anomalous size effect on strength below a critical size, as observed in the present study from both experimental and simulation results.

Specifically, compositional heterogeneity is considered to "roughen" the landscape for the dislocation motion. This can be correlated with an increase in the variation of the local stacking-fault energy (SFE) in each atomic plane. In large samples, the SFE on any atomic plane is averaged over numerous domains with low and high values of the local SFE so that the SFE of an atomic plane is consistent with the overall SFE of the material; it, therefore, is independent of the sample size. However, when the sample size decreases, the number of variations in the local SFEs is not statistically significant for the average value of the SFE to be

constant for any given plane. Accordingly, the distribution of SFE values for different planes broadens as the size of the plane is reduced, even though the average value across all planes remains roughly constant (Supplementary Fig. 2) (see also ref. [23]).

Considering that the SFE markedly affects the stress required to move the dislocations (as demonstrated in Supplementary Fig. 3), dislocation slip behavior should also vary on different parallel slip planes and be dependent on sample size. Such observations suggest that reducing the sample size would introduce more "weak planes", i.e., planes with low SFE, or low lattice friction. Therefore, such heterogeneous planar characteristics of HEAs primarily result in the sample size-dependent softening which is shown in Fig. 4d.

Furthermore, the value of $L_{critical}$ for the yield stress in HEAs can be tuned by changing the compositional heterogeneity. The MD results demonstrate that the annealed samples exhibit a larger critical size, i.e., $L_{critical}$ of ~28 nm, than the random solid-solution (RSS) configurations, where $L_{critical}$ of ~12 nm, as shown in Fig. 4c. We interpret this result as arising from the larger spatial extent of the compositional heterogeneities in the annealed sample (~1.5 nm) versus that in the random solid-solution samples (~0.56 nm) (Supplementary Fig. 4); the larger $L_{hetero}$ would shift the distribution of heterogeneity-induced softening to a higher length-scale, thus resulting in a larger critical size, $L_{critical}$, for the yield stress in HEAs.

To explore this interpretation further, we designed a simulation cell that displays clustering (compositional variations) on the scale of ~1 nm but with no chemical short-range order within the cells (Supplementary Fig. 5). As observed, the value of $L_{critical}$ of the yield stress for these artificially constructed CrCoNi alloys is also much larger than that of random solid-solution sample.

It should be noted the value of $L_{critical}$ (i.e., the size below which the samples display decreasing strength with decreasing size) for the eight-element FeCoNiCuPdIrPtAu nanoparticles that were investigated experimentally was ~180 nm, which is much larger than the values of $L_{critical}$ of 12 to 28 nm that were obtained for the MD-simulated three-component CrCoNi model alloys (Fig. 4c). Such a discrepancy can be attributed to the more pronounced compositional heterogeneity in the eight-element FeCoNiCuPdIrPtAu HEA, due to the nature of the interatomic interactions between the underlying alloying elements, which give rise to compositional variations with a length-scale ($L_{hetero}$) exceeding 10 nm (Fig. 1c).

Based on the results from combined experimental and computational simulation studies, we conclude that compositional heterogeneity plays an important role in affecting the mechanical

properties of HEAs. The effect of compositional heterogeneity on the yield stress for the experimental samples studied has a critical length-scale of at least tens of nanometers, which is much larger than that for random solid-solution strengthening. Below this critical size, the energy barrier for dislocation glide is highly influenced by the individual local chemical arrangements on specific slip planes; accordingly, the strength of the alloy can vary due to the heterogeneous distribution of planar properties. Such a size effect modified by chemical heterogeneity should commonly exist in complex alloys with a heterogeneous distribution of elements; its characteristic length-scale and effect on the yield strength and work hardening ability are sensitive to the composition and thermal processing of the alloys. Our results not only reveal a size-dependent yield strength and the role of compositional heterogeneity in influencing the mechanical properties of HEAs, but further provides insight into the design of concentrated solid solutions since the characteristic length-scale of this mechanism can be readily approached through control of composition and thermal history.

## Methods

**Sample preparation**. The HEA nanoparticles were prepared with the FMBP (Fast Moving Bed Pyrolysis) strategy[40]. Firstly, the mixed metal chloride salts precursors ($FeCl_3$, $CoCl_2$, $NiCl_2$, $CuCl_2$, $HAuCl_4$, $H_2PtCl_6$, $PdCl_2$, $IrCl_3$) coordinated with 1,10-Phenanthroline (Phen) were loaded on graphene oxide (GO) by the wet impregnation method; then, the aqueous solution was evaporated to dryness at 323 K in an ultrasound system. Subsequently, nanoparticles were produced by reductive pyrolysis at 923 K using the FMBP method. This entailed preparing metal precursors which were loaded on GO; these, in turn, were evenly placed in a quartz boat, initially 200 mm away from the heating region of the furnace. After changing the air with argon, the temperature of the furnace was increased to 923 K at a heating rate of 10 K·min$^{-1}$. The quartz boat supporting the samples was then pushed into the center of the hot zone at a speed of 200 mm·s$^{-1}$ for FMBP. (The quartz boat could reach the center of the hot zone within ~1 s). After annealing for 120 min at 923 K, the furnace was cooled naturally to room temperature to obtain the different sized $Fe_{18}Co_{16}Ni_8Cu_{20}Pd_{12.5}Ir_{1.5}Pt_{10}Au_{14}$ nanoparticles.

The particles were preserved *in vacuo* in a drying chamber to avoid oxidation. Prior to further characterization and in situ testing, they were ultrasonic dispersed in ethyl alcohol for 60 min to remove the supporting GO.

The spatial distribution and the relative atomic ratio of eight elements in the particles was studied using an aberration-corrected transmission electron microscope equipped with a large detecting angle Chemi-stem.

**In situ TEM compression testing**. The particles were compressed using in situ transmission electron microscopy (TEM) along crystallographic directions within 15 to 30 degrees of the <111> direction. In situ TEM mechanical tests on particles were performed using a Hysitron PI95 nanoindenter inside an FEI Tecnai G2 F20 TEM under a low current intensity of ~$1.8 \times 10^{-2}$ A·cm$^{-2}$, as previously reported. Particles with diameters varying from ~50 to 250 nm were ultrasonic dispersed in ethyl alcohol for 30 min and dropped at a sharp silicon wedge (thermal conductivity 149 W·m$^{-1}$K$^{-1}$) with a platform on the top so that the nanoparticles became attached by the intermolecular force. The wedge was pasted on the sample stage normal to the electron beam, and the top of the wedge was parallel to the front of the nanoindenter tip, which was flat and made of diamond (with thermal conductivity of 3320 W·m$^{-1}$K$^{-1}$), and controlled by a piezoceramic actuator behind it. The front of the tip was 2 μm wide in order to ensure each compression test only involved one single particle. The relative positions of the wedge and the tip were adjusted to ensure the particles were perpendicularly compressed. Compression testing was performed under displacement control (2 nm·s$^{-1}$) to better observe any transient phenomena[46]. The load *vs.* displacement data were simultaneously collected while the indentation process was recorded under a bright field mode of view.

**Stress calculations**. Engineering stress values were derived from the continuously recorded force values and the displacement of the nanoindenter tip (reduction-in-height). Since the deformation in this study was rather severe and the particles were more barrel-shaped, stress values were calculated using the cylinder model[47]. In this way, the particles during deformation were considered as cylinders. The height of cylinder equals to the distance between the wedge and nanoindenter, and the radius of the area that supports the load was estimated by setting the original volume of the sphere equal to the current column of the cylinder, which was

taken as

$$a = \left( \frac{4r^3}{3(2r - \delta_t)} \right)^{1/2} \tag{1}$$

where $r$ is the original radius of the particle, $\delta_t$ is the displacement of the nanoindenter tip at time $t$.

True stress data were calculated from recorded force values and radii measured from the extracted frames at specific time intervals. Diagrams using true stress values and real-time expansion in maximum width for certain points were used to compare the deformation in the different directions.

**MD simulation of the CrCoNi alloys**. Our MD simulations employed the recently developed EAM potential[16] for CrCoNi alloys to investigate the size dependence of the stacking-fault energies. All simulations were carried out using the LAMMPS[48]. The random solid-solution CrCoNi configurations, containing up to 1 million atoms, were then quenched to zero temperature and energy-minimized for subsequent calculation of the stacking fault energies. [110] oriented rhombic nanowires, with an aspect ratio of 2.5, were constructed with periodic boundary conditions in the axial direction and free surfaces in all other directions. Different sized nanowires were deformed with a tensile strain rate of $10^8$ s$^{-1}$ at 300 K. A number of nanowires (5–500 samples) of each sample size were simulated to ensure the convergence of the yield stress. For each sample size, we initially randomly distributed the atoms of each type into the lattice and selected those samples with $\sum \alpha_{ij,k} < 0.05$ for further investigation, where $\alpha_{ij,k}$ is the Warren–Cowley coefficient[49] between $i$ and $j$ types within the $k$th nearest neighbor shells ($k = [1,3,]$). The hybrid Molecular Dynamics and Monte Carlo simulations, under the variance-constrained semi-grand-canonical ensemble, were carried out to obtain the equilibrium configurations at different annealing temperatures[16]. The variance parameter κ used in our simulations was 1000. The artificially constructed CrCoNi alloy sample (in Supplementary Fig. 5) was designed to represent a supercell composed of an array of smaller unit cells with dimensions of ~$1 \times 1 \times 1$ nm$^3$. The concentration of each cell was randomly assigned for $(CrCo)_{1-x}Ni_x$, where $0 < x < 2/3$, in such a way that the average composition corresponds to Cr:Co:Ni = 1:1:1. In so doing, we developed a simulation cell to display clustering (compositional variations) on the scale of ~ 1 nm but with no chemical short-range order within the cells (Supplementary Fig. 5).

## Data availability

All data generated or analyzed during this study are included in the published article and are available from the corresponding authors upon request.

## Code availability

All related code are available from the corresponding authors on request.

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

## Acknowledgements
Q.Y. was funded by National Key Research and Development Program of China [grant numbers 2017YFA0208200], National Natural Science Foundation of China [grant numbers 51671168] and [grant numbers 51871197]. J.D. acknowledges support from National Natural Science Foundation of China (12004294), National Youth Talents Program and the HPC platform of Xi'an Jiaotong University. S.Y., M.A., and R.O.R. were supported by the Damage Tolerance in Structural Materials program (KC13) at the Lawrence Berkeley National Laboratory, funded by the U.S. Department of Energy, Office of Science, Office of Basic Energy Sciences, Materials Sciences and Engineering Division, under Contract No. DE-AC02-05CH11231.

## Author contributions
Q.Y. designed the research. Q.Y. and J.Y. performed TEM and in situ experiments. J.D. and S.Y. conducted data analysis and modeling. Q.Y., J.Y., J.D., M.A., and R.O.R. wrote the manuscript. All authors contributed to the discussion and revision of the paper.

## Competing interests
The authors declare no competing interests.
