## [Peer Review File · Nature Communications]

Title: Anomalous size effect on yield strength enabled by compositional heterogeneity in high-entropy alloy nanoparticlesEditorial Note: This manuscript has been previously reviewed at another journal that is not operating a transparent peer review scheme. This document only contains reviewer comments and rebuttal letters for versions considered at *Nature Communications*.

REVIEWER COMMENTS

Reviewer #3 (Remarks to the Author):

The authors have successfully addressed most of my previous concerns in this revised manuscript and have improved its quality. The manuscript is suited for publication in Nature Communications.

Reviewer #4 (Remarks to the Author):

This manuscript examines the size effect on the strength differences of a FeCoNiCuPdIrPtAu HEA nanoparticle system subjected to the size effects. A transition region from the size-dependent strengthening to inverse size-dependent relationship is found. The mechanical behavior of the FeCoNiCuPdIrPtAu HEA nanoparticle is observed by HADDF video, which is not trivial. The authors systematically design and develop different sizes of the samples for this specific HEA system. Moreover, the authors apply MD to simulate the size effect. The reported HEA system shows the strength transition. A hypothetical model is proposed to justify the rationality of the transition. It does contain a discussion of the reported effects. Some of the ideas presented will certainly stimulate discussions. The current submission will activate research in this field. There are however few general comments below concerning about the convenience for the readers to understand the experimental details of this work. Very minor but could perhaps help the audience.

1. Size effect on the HEAs has been reported, such as the grain size on fatigue behavior and the others. For example, Li et al.'s "Machine Learning Approach to Design High Entropy Alloys with Heterogeneous Grain Structures" on MMTA (2021) <https://link.springer.com/article/10.1007/s11661-020-06099-z>, Luo et al.'s "Grain-size-dependent microstructure effects on cyclic deformation mechanisms in CoCrFeMnNi high-entropy-alloys" on Scripta (2022) (<https://www.sciencedirect.com/science/article/abs/pii/S1359646221007375>), and Huang et al.'s "Machine-learning and high-throughput studies for high-entropy materials" on the Fig. 26, MSER (2022) (<https://www.sciencedirect.com/science/article/abs/pii/S0927796X21000401>). There is also transition region from the Hall–Petch to inverse Hall–Petch relationship for the HEAs. However, in this work, the effects on strengths shown in Fig 3 (lines 204~206, p.11) look greater than the aforementioned grain size effects on the HEAs.

1-1. Would the authors please use an additional table (or a figure) to specify the strengths differences (ex; how much percentage differences comparing the lowest to the highest strengths as a function of

each size) for this transition? The current 3-D columns in Fig. 3 are not clear to compare the absolute values of the stresses. It is also not easy to read the data for the red, green, blue, and cyan columns, respectively.

1-2. If the authors may consider summarizing the differences in strengths in this work, the reported effect can be quantitatively compared with the other effects shown in the HEAs and demonstrate the significance of the reported phenomenon. For example, in Liu et al's "Nanoprecipitate-Strengthened High-Entropy Alloys" (2021) *Advanced Science* (<https://onlinelibrary.wiley.com/doi/full/10.1002/advs.202100870>), their column bars shown in Fig. 6 demonstrate the significance/contribution of the Nanoprecipitate-Strengthen effect.

2. In the "Response to Reviewers' Comments Letter", there were concerns regarding the surface diffusion and the diffusion-mediated deformation mechanism, especially for the surface-to-volume ratio in nano structure.

2-1. Would the authors please give more details of the experimental conditions to rule out those concerns?

2-2. For example, with the continuous in-situ measurements, how much energy might be induced by the electron microscopy to the samples?

2-3. With the limited contacting area, how much heat would be conducted from the samples to the surroundings?

2-4. With this sample arrangement, how much heat might be accumulated on the samples during the measurements?

2-5. What could be the sample temperature when the measurements were made?

2-6. With different sizes, would the conduction rate of the heat be different for various samples?

2-7. For the sample temperature estimations, would the authors please provide the formula and associated values?

2-8. From the 5 videos provided, the shortest time of the measurement is 10 seconds (Deformation process of the 80nm-sized particle). The longest time of the measurement is one minute and 39 seconds (Deformation process of the particle with twins): would the authors please provide the strain rates for each test?

2-9. For the strain rate calculation, would the authors please provide the formula and associated values?

2-10. Would the authors please provide both the strain rates and the homologous temperature (T/T_m) for each test?

3. In the "Response to Reviewers' Comments Letter", there were concerns regarding the composition of the MD is different than that of the measured samples. Since the experimental results per se is significant, removing the MD part to the Supplementary may be an alternative (not mandatory).

Response to Reviewers' Comments Letter

Anomalous size effect on yield strength enabled by compositional heterogeneity in high-entropy alloy nanoparticles

(Manuscript # NCOMMS-21-50061-T)

We sincerely thank both reviewers for their exceptionally comprehensive and insightful review comments. We have attempted to revise the manuscript directly along the lines of their suggestions. We truly appreciate their efforts which we believe has made the manuscript so much better.

Reviewer #3 (Remarks to the Author):

The authors have successfully addressed most of my previous concerns in this revised manuscript and have improved its quality. The manuscript is suited for publication in Nature Communications.

Reviewer #4 (Remarks to the Author):

This manuscript examines the size effect on the strength differences of a FeCoNiCuPdIrPtAu HEA nanoparticle system subjected to the size effects. A transition region from the size-dependent strengthening to inverse size-dependent relationship is found. The mechanical behavior of the FeCoNiCuPdIrPtAu HEA nanoparticle is observed by HADDF video, which is not trivial. The authors systematically design and develop different sizes of the samples for this specific HEA system. Moreover, the authors apply MD to simulate the size effect. The reported HEA system shows the strength transition. A hypothetical model is proposed to justify the rationality of the transition. It does contain a discussion of the reported effects. Some of the ideas presented will certainly stimulate discussions. The current submission will activate research in this field. There are however few general comments below concerning about the convenience for the readers to understand the experimental details of this work. Very minor but could perhaps help the audience.

Response: We thank the reviewer's reading of our manuscripts and his/her constructive comments. We have carefully considered this advice and revised the manuscript directly along the lines suggested by these comments; our point-to-point response is as follows.

1. Size effect on the HEAs has been reported, such as the grain size on fatigue behavior and the others. For example, Li et al.'s "Machine Learning Approach to Design High Entropy Alloys with Heterogeneous Grain Structures" on MMTA (2021) (<https://link.springer.com/article/10.1007/s11661-020-06099-z>), Luo et al.'s "Grain-size-dependent microstructure effects on cyclic deformation mechanisms in CoCrFeMnNi high-entropy-alloys" on Scripta (2022) (<https://www.sciencedirect.com/science/article/abs/pii/S1359646221007375>), and Huang et al.'s "Machine-learning and high-throughput studies for high-entropy materials" on the Fig. 26, MSER (2022) (<https://www.sciencedirect.com/science/article/abs/pii/S0927796X21000401>). There is also

transition region from the Hall–Petch to inverse Hall–Petch relationship for the HEAs. However, in this work, the effects on strengths shown in Fig 3 (lines 204~206, p.11) look greater than the aforementioned grain size effects on the HEAs.

Response: We thank the reviewer for providing these reports and we found some of them were actually in support of our conclusions. We have added them as our references in the revised manuscript as references 37-39.

Li et al. reported the Hall-Petch to inverse Hall-Petch transition in FeCoCrNi HEAs at 38.4 nm through machine learning. But in this study, the authors assumed that the various elements were distributed uniformly; they accordingly, attributed the transition to the effect of grain boundary strengthening, which is not applicable in our study, where we used single crystal nanoparticles with a heterogeneous local chemical distribution and naturally no grain boundaries. Luo et al. investigated the effect of grain boundary hardening on the low-cycle fatigue properties. Specifically, in a fine-grained CoCrFeMnNi alloy, they found that the higher density of grain boundaries provides more impedance to dislocation motion resulting in higher flow stresses. Additionally, the consequent grain boundary assisted, wavy slip-driven, subgrain structures were beneficial to fatigue lifetimes. Figure 26 in Huang et al provided a comparison between the ML-designed HEA with heterogeneous grain structures and mono-dispersed grain size.

However, in our report, we focused on the single crystal HEA nanoparticles (without grain boundaries) that are fundamentally different with that studied in the previous literature. We quantified, we believe for the first time, the effect of compositional heterogeneity on the yield strength and strengthening at its intrinsic correlation length. Our hypothesis was that compositional heterogeneity would influence dislocation behavior as the correlation length of this heterogeneity is much larger than the intrinsic spatial length-scale of composition variation. This implies an additional strengthening mechanism in materials with such composition heterogeneity. How significant this mechanism is and how large its correlation length is in tuning the materials properties is sensitive to the degree of heterogeneity which is clearly related to the composition. So we believe that our results can provide insight into new understanding of the deformation of complex alloys.

1-1. Would the authors please use an additional table (or a figure) to specify the strengths differences (ex; how much percentage differences comparing the lowest to the highest strengths as a function of each size) for this transition? The current 3-D columns in Fig. 3 are not clear to compare the absolute values of the stresses. It is also not easy to read the data for the red, green, blue, and cyan columns, respectively.

Response: We thank the reviewer for this usual advice. We've added a table, listing the yield stress and the hardening rate at the strain of 40% (calculated as the percentage differences comparing the stress at strain of 40% to that at the yield point). This table shows the transition of the “smaller-is-stronger” to the “smaller-is-weaker” phenomena with the reduction in particle size together with the change in work-hardening ability. The yield stresses decrease as the sample size is reduced below particles sizes of ~180 nm in the form of a “smaller-is-weaker” trend. Additionally, the work-hardening ability markedly deteriorates, from 48% to around 0%, after the “smaller-is-stronger” to “smaller is weaker” transition.

Table R1 The yield stress and strain hardening ratio of different sized samples

Particle diameter	Yield stress	Strain hardening ratio (at 40% strain)
80 nm	0.8 GPa	18.7 %
140 nm	1.08 GPa	-1 %
180 nm	1.25 GPa	0 %
200 nm	1 GPa	48 %
260 nm	0.78 GPa	47.4 %

1-2. If the authors may consider summarizing the differences in strengths in this work, the reported effect can be quantitatively compared with the other effects shown in the HEAs and demonstrate the significance of the reported phenomenon. For example, in Liu et al's "Nanoprecipitate-Strengthened High-Entropy Alloys" (2021) *Advanced Science* (<https://onlinelibrary.wiley.com/doi/full/10.1002/advs.202100870>), their column bars shown in Fig. 6 demonstrate the significance/contribution of the Nanoprecipitate-Strengthen effect.

Response: We thank the reviewer's for the good suggestion. In this research, we proposed that the chemical heterogeneity would lead to significant softening effect in small-scaled samples and compete with the classical "smaller is stronger" effect, resulting in an anomalous "smaller is weaker" size effect below a critical size. Further we proposed a simplified model incorporating the softening effect of yield strength in HEAs, as below:

$$\sigma = \sigma_0 + k * d^{-n} - b * \exp\left(-\frac{d}{\alpha * L_{hetero}}\right),$$

where the first two terms were previously used to describe the smaller-is-stronger effect for the sample size d (see the illustration in main text) and the third term is for the heterogeneity-induced softening; L_{hetero} is the characteristic length of heterogeneity as discussed in main text. b and α are two constant factors. However, the empirical parameters σ_0 , k , n , b and α cannot be determined with the limited data available to date and quantitative comparison of these two factors needs more future study.

However, we modified Fig. 4d (here presented as Fig.R1) in the similar form to Fig. 6 in the above reference such that we hope the competition relation of the two factors is illustrated more straightforwardly.

We have added this paper as our reference in the revised manuscript as reference 8.

Fig. R1 The schematic description for the size effect in high-entropy alloys, combining the classical strengthening effect for small-volume metals/alloys and the unique heterogeneity-induced softening for HEAs.

2. In the “Response to Reviewers’ Comments Letter”, there were concerns regarding the surface diffusion and the diffusion-mediated deformation mechanism, especially for the surface-to-volume ratio in nano structure.

2-1. Would the authors please give more details of the experimental conditions to rule out those concerns?

Response: We thank the reviewer for pointing this out. We have added more details in the Methods section including, but not confined, to the current intensity ($\sim 1.8 \times 10^{-2} \text{ Acm}^{-2}$) and the thermal conductivity of the diamond tip and silicon wedge ($149 \text{ Wm}^{-1}\text{K}^{-1}$ and $3320 \text{ Wm}^{-1}\text{K}^{-1}$, respectively¹), which are important to appreciate the minimal effect of surface diffusion.

The low current intensity in the experiment is much lower than the intensity for normal HRTEM observation ($5\text{-}8 \text{ A/cm}^2$), and is much lower than those used e-beam for heating and welding ($10^5\text{-}10^6 \text{ A/cm}^2$), which would locally heat samples to their melting points.

Additionally, the diamond tip and silicon wedge adopted in our experiments are ideal thermal conductors (with thermal conductivities of $149 \text{ Wm}^{-1}\text{K}^{-1}$ and $3320 \text{ Wm}^{-1}\text{K}^{-1}$, respectively¹), and thus would minimize heat concentration at the contacting surfaces.

In the following parts, we provide a thorough discussion of the heat transport during the experiment to clearly demonstrate that the beam-heating effect would not be a significant factor during the deformation process.

2-2. For example, with the continuous in-situ measurements, how much energy might be induced by the electron micrography to the samples?

Response: The energy that the sample obtained during the experiment is mainly due to the energy loss of the electrons caused by collision and bremsstrahlung. According to the Bethe-Bloch equation²,

$$Q = -\frac{d\bar{E}}{dx} = \frac{2\pi Z\rho(e^2/4\pi\epsilon_0)^2}{mv^2} \left\{ \ln \left[\frac{E(E+mc^2)\beta^2}{2I_e^2 mc^2} \right] + (1-\beta^2) - (2 - \sqrt{1-\beta^2} - 1 + \beta^2) \ln 2 + \frac{1}{8}(1 - \right.$$

$$\sqrt{1 - \beta^2}\}^2,$$

where Z is the atomic number of the target element, and ρ is the atomic density. Here $\beta = \frac{v}{c}$, with c equal to the speed of light and v is the electron velocity. ϵ_0 is the vacuum permittivity, and e and m the electron charge and rest mass. E is the electron energy, and I_e is the average excitation energy for electrons in the target. Adopting the average value of the alloying element, the energy loss per length of the 200 kV-accelerated electrons in our HEA particles, namely the stopping power Q , is calculated to be 1.907 eV/nm. The e-beam energy is transferred to the sample at an energy density rate of

$$H = \frac{QJ}{e} = 3.43 \times 10^7 \text{ Wm}^{-3},$$

where J is the electron current density ($1.8 \times 10^{-2} \text{ Acm}^{-2}$) and e is the elementary charge ($1.6 \times 10^{-19} \text{ C}$).

Given above analysis, the energy transfer power P can be calculated from the volume V of the particle by:

$$P=HV.$$

Note that the result of the particles with a diameter over 150 nm might be larger than the real situation since the decay distance of the electrons cannot reach such a dimension, which would result in the poor contrast in the middle of the particle in the TEM images (Fig.1). Here we take the extreme situation, assuming all particles (with diameter from 80 to 260 nm) are electron transparent and calculate the energy transfer. The results are listed in Table R2.

Table R2 the energy transfer power of particles with different sizes

Diameter (nm)	Energy transfer power P (10^{-11} W)
80	9.19
140	49.29
180	104.77
200	143.71
260	315.73

2-3. With the limited contacting area, how much heat would be conducted from the samples to the surroundings?

Response: During the *in situ* experiment, the particle is in contact with the silicon wedge and diamond tip, which are of infinite size compared to the particles. The thermal conductivities of the silicon and diamond are $149 \text{ W}\cdot\text{m}^{-1}\text{K}^{-1}$ and $3320 \text{ W}\cdot\text{m}^{-1}\text{K}^{-1}$, respectively, and thus can be deemed as ideal cold sinks. Therefore, the heat accumulation in the sample is determined by the ability of thermal transportation of the interface (interface thermal conductance), which is widely discussed, ranging from $<0.01 \text{ MW}\cdot\text{m}^{-2}\text{K}^{-2}$ to $1000 \text{ MW}\cdot\text{m}^{-2}\text{K}^{-1}$ depending on the contacting situation³⁻⁵. In our study, the asperity of the nanoscale contacting interface would be easily eliminated with the application of high pressure because of plastic fit¹, which yields a high interface thermal conductivity close to that of the epitaxy or deposition. In fact, our Supplementary Movie 2 shows that considerable force is needed to pull the sample and probe apart after compression, which would overcome the strong adhesion applied at the interface. There have been some reports on the contact

thermal conductance in such situations, including that of the Pt-Rh probe of scanning thermal microscope and various sample surfaces⁶, adhered microcantilevers on the substrate⁷, as well as the contact between wafer-like Al/Si solid surfaces under ~10 MPa pressure⁸. Values of the thermal conductivity G in these cases are all reported to be larger than $1 \text{ MW}\cdot\text{m}^{-2}\text{K}^{-1}$. We thus take $G=1 \text{ MW}\cdot\text{m}^{-2}\text{K}^{-1}$ for following analysis, considering the intimate contact between the particle and the silicon wedge and diamond tip¹. This yields a temperature difference at the interface of:

$$\Delta T = \frac{HV}{GA},$$

where V is the sample volume and A is the total contact area, which is measured from the frames extracted from the videos. Taking the contacting area measured from the video, values of the temperature difference ΔT between the contact interface during the deformation process are listed in Table R2. The largest temperature difference come up at the beginning of the compression, where the contacting area is smallest. During the compression, the contacting area increase rapidly and temperature differences at the interface further decrease. In this research, the maximum of ΔT is ~20.11 K (note the real temperature difference is even smaller because of the limited electron passing distance and a smaller energy transfer power P discussed previously), which is negligible compared to its melting temperature (over 1000 K).

Table R2 The temperature difference ΔT during deformation process (unit: K)

Particle diameter \ Strain	0%	10%	20%	30%
80 nm	14.65	0.09	0.065	0.037
140 nm	19.62	0.13	0.064	0.049
180 nm	18.53	0.018	0.067	0.034
200 nm	14.3	0.019	0.085	0.028
260 nm	20.11	0.021	0.096	0.042

2-4. With this sample arrangement, how much heat might be accumulated on the samples during the measurements?

Response: Based on above discussion, we know that the heat accumulated at the interface is negligible. However, the temperature inside the sample is not uniform and the temperature at the center of the sample is slightly higher than that of the interface due to the rather low thermal conductivity of the HEA ($k\sim 30 \text{ Wm}^{-1}\text{K}^{-1}$)^{9,10}. At equilibrium (which is evident from the minimal change in the level of the compression stress which would otherwise decrease if heat continuously accumulate inside the particle¹), the temperature field inside the sample satisfies

$$k \nabla_{\vec{r}}^2 T = -H,$$

while at the hottest point (center of the particle in this case) the temperature gradient is zero. We then would have the temperature gradient $\nabla_{\vec{r}}^2 T = -1.14 \times 10^{10} \text{ K}\cdot\text{m}^{-2}$ and the temperature difference can be estimated with the sample dimension, to be small as 10^{-3} K , which means during the compression process, the whole sample is as cold as the substrate and probe tip and the heat accumulation is negligible.

2-5. What could be the sample temperature when the measurements were made?

Response: According to previous discussion, the sample temperature before the compression is

slightly higher (about 10~20 K) than the environment; during the compression process the sample temperatures is kept same with the environment (room temperature).

2-6. With different sizes, would the conduction rate of the heat be different for various samples?

Response: For the same material under same experiment conditions, the stopping power (the energy loss per distance travelled of swift electrons), and thus the energy density rate for e-beam energy transfer, together with the thermal conductivity of the sample and contact interface, all remain constant. The differences in heat accumulation are mainly caused by the difference in sample volumes and the sizes of contact area between the sample and the diamond tip and silicon wedge, which have been taken into consideration into the calculation of ΔT in response for 2-3. These results show that the differences in sample size only cause minor differences in ΔT .

2-7. For the sample temperature estimations, would the authors please provide the formula and associated values?

Response: The estimation of sample temperature involves the Bethe-Bloch formula to calculate the energy loss of electrons while travelling through the sample (response for 2-2), then adopted classical heat transfer theory to calculate the temperature difference between the contact interface (response for 2-3) and the temperature field inside the sample (response for 2-4). The associated values are provided accordingly.

2-8. From the 5 videos provided, the shortest time of the measurement is 10 seconds (Deformation process of the 80 nm-sized particle). The longest time of the measurement is one minute and 39 seconds (Deformation process of the particle with twins): would the authors please provide the strain rates for each test?

Response: As described in Method in the manuscript, the compression tests were performed under displacement control, by which rich nanomechanical phenomena would show up¹¹. The moving speed of the tip was set as 2 nm/s in all experiments, eliminating the system error of different setting conditions. Under such circumstances, the strain rate (defined as the derivative of the contraction in height with respect to time) at the order of magnitude of $10^{-3}\sim 10^{-2}/s$, which belongs to the quasi-statistic testing condition, under which the deformation behavior and yield stress show low strain-rate dependence¹². However, the videos include some other processes like the movement of the tip from some distance away from the sample as well as the unloading process, and thus show some differences in the length. The strain-time relations are plotted in Fig. R2, with the strain rates marked.

Fig. R2 The strain-time relation of each compression process (with strain rates marked over the curves)

2-9. For the strain rate calculation, would the authors please provide the formula and associated values?

Response: In this research, the stress is applied along the particle height direction and strain ϵ is defined as the ratio of the contraction in height H_0-H_t (recorded as the displacement of the tip) of the initial height H_0 , in the form of $\epsilon_t = \frac{H_0-H_t}{H_0}$. Additionally, the strain rate is calculated as the derivative of the contraction in height with respect to time. Since the tip forward at fixed speed, the strain rate in each compression test remains constant, and equals the slope of the strain-time curve in Fig. R1. Based on these calculations, the estimated strain rates for the compression tests of 80, 140, 180, 200 and 260 nm-sized particles are, respectively, 1.9×10^{-2} , 1.0×10^{-2} , 7.1×10^{-3} , 5.3×10^{-3} and 4.1×10^{-3} .

2-10. Would the authors please provide both the strain rates and the homologous temperature (T/T_m) for each test?

Response: The strain rates and the homologous temperature for each test is listed in Table R3 as follows.

Table R3 The strain rates of the compression and the homologous temperature of the particles for each test (room temperature as 293 K)

Particle diameter	Strain rate	Temperature (before test)	Temperature (during test)
80 nm	1.9×10^{-2}	307.65 K	293.09 K
140 nm	1.0×10^{-2}	312.62 K	293.13 K
180 nm	7.1×10^{-3}	311.53 K	293.01 K
200 nm	5.3×10^{-3}	307.3 K	293.01 K
260 nm	4.1×10^{-3}	313.11 K	293.01 K

3. In the “Response to Reviewers’ Comments Letter”, there were concerns regarding the

composition of the MD is different than that of the measured samples. Since the experimental results per se is significant, removing the MD part to the Supplementary may be an alternative (not mandatory).

Response: We thank the reviewer for the advice. We agree that there is material discrepancy between CrCoNi alloy (atomistic simulated) and 8-element HEA (by experiment), although both are single-phase, *fcc* concentrated solid solutions. This situation is not unusual for studies of modeling and simulations. Actually, in most cases of modeling, one cannot fully reproduce the experimental conditions and simulate the exact same materials. Instead, the most important value of modeling/simulation is to extract the underlying mechanism and physics from a model system, which can help to clarify the origins of experimental observations. This is a challenge but the strength of materials modeling and simulation.

In this research, the MD simulation not only show results of NiCoCr HEAs that are similar with the experiments, but also provided evidence of the effect of such chemical heterogeneity in the distribution of SFE and therefore the behavior of dislocation motion (see L251-274 and Extended Fig.2-5), which would greatly help the reader to understand our argument. Thus, we have elected to keep this part in the manuscript; we hope that you concur.

Reference

- 1 Zheng, K. *et al.* Electron-beam-assisted superplastic shaping of nanoscale amorphous silica. *Nature Communications* **1**, 24 (2010).
- 2 Jencic, I., Bench, M., Robertson, I. & Kirk, M. Electron-beam-induced crystallization of isolated amorphous regions in Si, Ge, GaP, and GaAs. *Journal of Applied Physics* **78**, 974-982 (1995).
- 3 Lyeo, H.-K. & Cahill, D. G. Thermal conductance of interfaces between highly dissimilar materials. *Physical Review B* **73**, 144301 (2006).
- 4 Fletcher, L. Recent developments in contact conductance heat transfer. (1988).
- 5 Aikawa, T. & Winer, W. O. Thermal contact conductance across Si₃N₄—Si₃N₄ contact. *Wear* **177**, 25-32 (1994).
- 6 Lefèvre, S. & Volz, S. 3 ω -scanning thermal microscope. *Review of scientific instruments* **76**, 033701 (2005).
- 7 Huxtable, S. T., Cahill, D. G. & Phinney, L. M. Thermal contact conductance of adhered microcantilevers. *Journal of applied physics* **95**, 2102-2108 (2004).
- 8 Ohson, Y., Wu, G., Dryden, J., Zok, F. & Majumdar, A. Optical measurement of thermal contact conductance between wafer-like thin solid samples. (1999).
- 9 Caro, M., Béland, L. K., Samolyuk, G. D., Stoller, R. E. & Caro, A. Lattice thermal conductivity of multi-component alloys. *Journal of Alloys and Compounds* **648**, 408-413 (2015).
- 10 Kush, L., Srivastava, S., Jaiswal, Y. & Srivastava, Y. Thermoelectric behaviour with high lattice thermal conductivity of Nickel base Ni₂CuCrFeAl_x (x= 0.5, 1.0, 1.5 and 2.5) high entropy alloys. *Materials Research Express* **7**, 035704 (2020).
- 11 Warren, O. L., Downs, S. A. & WYROBEK, T. J. Challenges and interesting observations associated with feedback-controlled nanoindentation. *Zeitschrift für Metallkunde* **95**, 287-296 (2004).
- 12 Kumar, N. *et al.* High strain-rate compressive deformation behavior of the Al_{0.1}CrFeCoNi high entropy alloy. *Materials & Design* **86**, 598-602 (2015).

REVIEWERS' COMMENTS

Reviewer #4 (Remarks to the Author):

The authors addressed all the questions seriously. There are just few typos shown on the "SUPPLEMENTARY INFORMATION" for revision.

1. Page 4th: line 107
2. Page 5th: line 109